# Functional Outcomes of Cochlear Implantation in Children with Bilateral Cochlear Nerve Aplasia

**DOI:** 10.3390/medicina58101474

**Published:** 2022-10-17

**Authors:** Goun Choe, Young Seok Kim, Seung-Ha Oh, Sang-Yeon Lee, Jun Ho Lee

**Affiliations:** 1Department of Otolaryngology-Head and Neck Surgery, Chungnam National University Sejong Hospital, Chungnam National University College of Medicine, Sejong 30099, Korea; 2Department of Otorhinolaryngology-Head and Neck Surgery, Seoul National University Hospital, Seoul National University College of Medicine, Seoul 03080, Korea; 3Sensory Organ Research Institute, Seoul National University Medical Research Center, Seoul 03087, Korea

**Keywords:** cochlear nerve aplasia, cochlear implantation, correction of hearing impairment, vestibulocochlear nerve diseases

## Abstract

*Background and Objectives*: Many otologists face a dilemma in the decision-making process of surgical management of patients with cochlear nerve (CN) aplasia. The goal of this study is to provide fresh evidence on cochlear implantation (CI) results in patients with CN aplasia. *Materials and Methods*: We scrutinized functional outcomes in 37 ears of 21 children with bilateral CN aplasia who underwent unilateral or bilateral CI based on cross-sectional and longitudinal assessments. *Results*: The Categories of Auditory Performance (CAP) scores gradually improved throughout the 3-year follow-up; however, variable outcomes existed between individuals. Specifically, 90% of recipients with a 1-year postoperative CAP score ≤1 could not achieve a CAP score over 1 even at 3-year postoperative evaluation, while the recipients with a 1-year postoperative CAP score >1 had improved auditory performance, and 72.7% of them were able to achieve a CAP score of 4 or higher. Meanwhile, intraoperative electrically evoked compound action potential was not correlated with postoperative CAP score. *Conclusions*: Our results further refine previous studies on the clinical feasibility of CI as the first treatment modality to elicit favorable auditory performance in children with CN aplasia. However, special attention should be paid to pediatric patients with an early postoperative CAP score ≤1 for identification of unsuccessful cochlear implants and switching to auditory brainstem implants.

## 1. Introduction

Cochlear nerve (CN) deficiency refers to a small or absent CN; it is clinically diagnosed using high-resolution imaging, and was first described by Casselman in 1997 [1]. The CN is considered ‘aplastic’ if it cannot be identified on temporal bone imaging, including oblique sagittal imaging; and ‘hypoplastic’ if it appears smaller than the facial nerve within the internal auditory canal (IAC) on oblique sagittal imaging [2]. Previous studies have demonstrated that CN deficiency, one of the major inner ear anomalies, is seen in approximately 1–5.3% of children with bilateral sensorineural hearing loss (SNHL) [3,4]. CN deficiency, especially CN aplasia, has traditionally been considered a contraindication for cochlear implantation (CI) because the outcomes of CI are largely associated with CN status [5,6,7], even in the severely malformed cochlea [8]. In most patients, the presence of the CN and modiolus can be confirmed using magnetic resonance imaging (MRI) [8]. However, there were limitations in cases with a very thin cochleovestibular nerve (CVN) [9], thus suggesting that CN aplasia on MRI is not an absolute contraindication for CI [6,10]. The possibility of the presence of a CN fiber should be considered even when preoperative evaluation points to apparent CN aplasia. Correspondingly, evidence of CN deficiency as a potential indication for CI has increased [3,4].

Several studies evaluating CI outcomes in children with CN deficiency have suggested that CN integrity should be considered in the decision-making process of surgical management. Theoretically, the higher the spiral ganglion neuron population, the better the CI outcome. This is in line with the results of previous studies that children with CN hypoplasia usually have similar or poorer performance than those with normal CN [11,12]. Meanwhile, children with CN aplasia are less likely to benefit from CI than those with CN hypoplasia [6,12]. However, variable CI outcomes exist in children with CN aplasia, resulting in a range of auditory performance, from awareness of environmental sounds to conversation without lipreading [13], implying that a subset of patients with CN aplasia can attain closed or open-set levels of speech perception following CI [6]. Considering the unpredictable outcomes of CI in patients with CN aplasia, it may be inappropriate to apply the same rehabilitation strategy to all patients, and CI should not be obligatorily pursued and maintained longitudinally. Indeed, some studies have shown that auditory brainstem implants (ABI) improve hearing in children with CN aplasia who had unsuccessful bilateral CI operations [6]. Specifically, Yousef et al. demonstrated better auditory perception and language development outcomes in the pediatric ABI group than in the CI group [2].

There is currently limited evidence on CI outcomes with apparent CN aplasia, preventing timely and appropriate auditory rehabilitation (e.g., choice between CI and ABI) in the context of the decision-making process of surgical management. Herein, we aimed to scrutinize functional outcomes in children with bilateral CN aplasia who underwent unilateral or bilateral CI using cross-sectional and longitudinal assessments. Based on our evidence, we suggest a prognostic indicator for determining candidacy for ABI after CI in children with bilateral CN aplasia. We believe that the results of this study further refine those of previous studies and suggest a potential guideline on how otologists should deal with CN aplasia in children in this era of customized auditory rehabilitation. 

## 2. Materials and Methods

### 2.1. Participants

We retrospectively reviewed the medical records of patients who underwent CI between July 2010 and June 2018 from the CI database of a single tertiary hospital. Among them, only individuals who met the following inclusion criteria were included: (1) bilateral CN aplasia diagnosed based on both IAC MRI (nonvisible CN on oblique sagittal imaging) and temporal bone computed tomography (absence of bony cochlear narrow canal) (Figure 1); (2) absence of auditory brainstem response to click stimuli at 90 dB with negative otoacoustic emissions in both ears; and (3) implantation by two experienced surgeons. Additionally, the exclusion criteria were as follows: (1) history of cochlear explantation or reimplantation; (2) severely malformed cochlea, including a common cavity, cochlear aplasia with dilated vestibule, and incomplete partition; (3) brain abnormalities observed during neuroradiological evaluation; and (4) syndromic deafness associated with neurodevelopmental delays, such as Charge syndrome and Noonan syndrome. Ultimately, 21 patients (37 ears, mean age 17.71 months) were included in this study. The Institutional Review Board approved this study, which was conducted in accordance with the tenets of the Declaration of Helsinki (IRB No. 2111-085-1273). Due to the retrospective design, informed consent was waived (or exempted) from the Institutional Review Board.

### 2.2. Audiological Evaluation

Auditory perception performance was assessed preoperatively and postoperatively according to eight categories using the Categories of Auditory Perception (CAP) scores, using a hierarchical scale from 0 to 7 for children’s developing auditory abilities. CAP is a validated and widely used parameter of auditory receptive ability worldwide [14]. Auditory performance was assessed at up to five time points during the 3-year follow-up period. All patients enrolled in this study completed the evaluation of CAP scores at each time point during the 3-year follow-up period.

### 2.3. Electrically Evoked Compound Action Potential and Mapping

Electrically evoked compound action potentials (ECAPs) from the intracochlear electrode were recorded in the operating room immediately after surgery. The positive response of ECAP was assessed in every channel for all patients using automatic telemetry programs intraoperatively, including neural response telemetry for cochlear neural response imaging for advanced bionics, and auditory response telemetry for MED-EL. As outlined in the guidelines [15], test conditions for neural response telemetry were set at a 250 Hz stimulation rate and 35 sweeps. The maximum stimulus was set at a current level of 255 CL. As the default setting for auditory response telemetry, the stimulation rate was 80 Hz, and the number of displayed responses was 5. The maximum charge was set to 35 charge units (qu), and the charge increase rate was 8.0 qu/s. For neural response imaging, the maximum stimulus was 400 current units. Mapping was performed throughout a three-year follow-up period, on the same schedule as the speech evaluation tests, including Categories of Auditory Performance (CAP). The average current unit threshold level (T Level) and maximum comfortable loudness level (C Level) were measured. The dynamic range was assessed as the calculated difference between the T and C levels. Mapping parameters, including sound coding, stimulation rate, pulse width, and input processing, were adjusted by an experienced audiologist.

### 2.4. Statistical Analysis

All analyses were performed using the R software package, version 3.3.2 (R Foundation for Statistical Computing, Vienna, Austria). Data were visualized using the R software package and GraphPad Prism 7.00 (GraphPad Software, San Diego, CA, USA). Data are presented as mean ± standard deviation (SD). Demographic and clinical variables were compared between groups using the Mann–Whitney U test and Fisher’s exact test, as appropriate. The Wilcoxon signed-rank test (matched) was used for the comparison of overall CAP scores between evaluation time points. Furthermore, Spearman’s correlation analysis was used to investigate the relationship between the ECAP response rate and postoperative CAP score. ANOVA and post hoc analysis were used for the comparison of T and C levels and dynamic range. Statistical significance was set at *p* < 0.05.

## 3. Results

### 3.1. Demographic and Clinical Characteristics

A total of 21 pediatric cochlear implantees (37 ears) with bilateral CN aplasia were included in the analysis, and all participants were followed up at all time points up to 3 years after surgery. CN aplasia was verified using both computed tomography and MRI (Figure 1). The demographic and clinical characteristics of the participants are presented in Table 1. Of these, 16 patients (76.2%) underwent simultaneous bilateral CI. The average age at the time of CI was 17.71 months (range: 10–37 months). Except for one patient who underwent surgery at 37 months of age, the majority of patients underwent surgery between 1 and 2 years of age (range, 10–28 months). Eight (38.1%) patients had inner ear anomalies other than CN aplasia, but not severely malformed cochleae, such as a common cavity (Appendix A). CI was performed via the round window approach in most patients, except for five patients who underwent CI via cochleostomy due to an invisible or inaccessible round window. Importantly, no major perioperative or postoperative complications were observed.

### 3.2. The Natural Course of Auditory Performance after Surgery

The overall CAP score improved significantly and gradually from 0.29 ± 0.55 to 2.67 ± 1.73 throughout the 3-year follow-up period (*p* < 0.001, Wilcoxon signed-rank test) (Figure 2). Specifically, a significant improvement in CAP scores between each time point, especially between preoperative and postoperative 1-year evaluation, was observed. However, none of the patients had a CAP score >5 even after a 3-year postoperative follow-up period. Of note, the CAP score differed significantly between individuals, indicating the importance of detailed exploration of the natural course of auditory performance depending on early-stage auditory development.

### 3.3. Prognostic Value of CAP Score at 1 Year

Our cohort was subdivided into four groups based on the CAP score at the early postoperative stage (i.e., 1-year evaluation time point), and we evaluated the natural course of auditory performance throughout the 3-year follow-up period (Figure 3). Three patients had a CAP score of 0 at 1 year postoperation. Among them, the CAP score of two patients slightly increased (up to 1), while the CAP score of the remaining patients remained steady (Figure 3A). Seven patients exhibited a CAP score of 1 at 1 year postoperation. Among them, the CAP score improved up to a value of 3 in only one patient and remained steady or dropped to 0 in six patients (Figure 3B). Patients with a CAP score >1 at 1 year postoperation tended to achieve better auditory performance over the 3-year follow-up period. Three of the five participants with a CAP score of 2 or 3 at 1 year postoperation eventually achieved a CAP score of 4 (Figure 3C). Of the four patients who scored 4 at 1 year post-CI, one scored 4 and two scored 5 at 3 years post-CI. However, in one patient (Pt 19), the CAP score dropped to 3 at 2 years post-CI and remained at 3 at the last follow-up (Figure 3D). Two patients with a CAP score of 5 at 1 year post-CI scored 4 or 5 at 3 years post-CI.

Stratified by CAP score ≤1 (*n* = 10) vs. CAP score >1 (*n* = 11) at 1 year postoperation, no significant differences in other confounding factors, such as age at CI, sex, laterality, insertion approach, and preoperative CAP scores, were noted (Table 2). However, CAP scores were significantly different between the two groups at all time points from preoperative evaluations (0 vs. 0.55 ± 0.66, *p* = 0.02, Mann–Whitney U test) to 3 years postoperation (1 ± 0.8 vs. 4.2 ± 0.6, *p* < 0.001, Mann–Whitney U test). Collectively, most patients with a CAP score ≤1 at 1 year postoperation remained steady or exhibited extremely poor auditory performance throughout the follow-up period, whereas most patients with a CAP score >1 at 1 year postoperation had markedly improved auditory performance, with a CAP score of at least 3.

### 3.4. Intraoperative ECAP

Of the 37 ears that underwent CI, intraoperative ECAP data were available for 34 ears. The positive ECAP ratio, referred to as the percentage of the number of electrodes exhibiting a positive response compared to the total number of electrodes, varied between ears, showing an average positive rate of 47.91 ± 36.03%. No correlation was observed between the positive ECAP ratio and postoperative CAP scores at each time point (Figure 4), although a weak tendency for greater auditory performance at 2 and 3 years postoperation was observed in ears with a positive ECAP ratio (12 months, R^2^ = 0.01, *p* = 0.69; 24 months, R^2^ = 0.14, *p* = 0.12; 36 months, R^2^ = 0.14, *p* = 0.11, respectively). These results imply that it would be difficult to predict the improvement of auditory performance postoperatively solely based on intraoperative electrophysiological testing using the positive ECAP ratio in children with CN aplasia.

### 3.5. Mapping

We evaluated the mapping data, including T and C levels, and dynamic range, when available. The average T and C levels and dynamic range did not significantly differ between the evaluation time points (C level, *p* = 0.99; T level, *p* = 0.82; dynamic range, *p* = 0.97, ANOVA and post hoc analysis) (Table 3).

## 4. Discussion

We presented the longitudinal auditory performance of CI in children with bilateral CN aplasia. Our study has some merits for the following reasons. This study included the largest number of patients with bilateral CN aplasia to explore the outcomes of CI based on cross-sectional and longitudinal audiological analyses. The overall CAP score showed a significant and gradual improvement from 0.29 ± 0.55 to 2.67 ± 1.73 throughout the 3-year follow-up period, suggesting that CI may elicit favorable auditory performance even in children with CN aplasia. However, improvement in auditory performance showed variability between individuals, and the benefit appeared to be predominant in patients with a CAP score >1 at 1 year postoperation. The results of this study may serve as a possible prognostic factor to aid in the decision-making process of surgical management of children with CN aplasia. Specifically, in pediatric cochlear implantees with a CAP score ≤1 at 1 year postoperation, transiting auditory rehabilitation from CI to ABI may be considered for better auditory performance. 

Since the first report proposed that CI could be applied to children with CN aplasia [16], very few studies on CI outcomes in patients with CN aplasia have been reported. Wu et al. [17] demonstrated significantly lower CAP scores in children with CN aplasia than in those with normal CNs matched for demographics. In addition, they reported that the patients in the aplasia group were all lower than the CAP score of 5 (understanding of common phrases without lipreading), which is in line with our results. Specifically, in this study, the most common CAP score after 3 years postoperatively was 4, possibly discriminating some speech without lipreading. Furthermore, Birman et al. demonstrated that approximately 50% of children with CN aplasia achieved some verbal understanding, as evidenced by a CAP score of 5 to 7. Meanwhile, Yousef et al. reported a mean 2-year postoperative CAP score of 1.29 in seven cases with CN deficiency [2]. Of these, five patients with CN aplasia scored 0, indicating no awareness of environmental sounds. The auditory performance of pediatric cochlear implantees with apparent CN aplasia presented in this study fell within the middle range based on reported studies in the literature. Although differences in methodologies, such as inclusion criteria and follow-up period, may lead to significant discrepancy, cross-sectional assessment and lack of evaluation time points would limit CI outcomes in patients with CN aplasia [2].

There is insufficient evidence on the criteria for determining the treatment modalities (CI vs. ABI) in children with CN aplasia. A recent study by Yousef et al. [2] compared and analyzed auditory performance after implantation in 14 patients with CN deficiency. Of these, seven patients underwent CI, and the other seven patients underwent ABI. Five patients in the CI group and all patients in the ABI group had bilateral CN aplasia. The mean CAP score at 2 years postoperation was 1.29 in the CI group and 2.87 in the ABI group, demonstrating better outcomes in the ABI group. However, not all studies in the literature fully support this, probably due to small sample sizes for statistical significance, confounding factors, or heterogeneous assessment of auditory and speech performance in patients with CN aplasia. Specifically, a meta-analysis demonstrated that among pediatric patients with CN deficiency, 25% (27/108) attained open-set speech perception and 34% (37/108) attained close-set speech perception after CI [6], suggesting that CI may serve as an initial treatment before ABI in children with CN deficiency. The rationale behind this could be the presence of residual CN fibers that were too hypoplastic to appear on MRI, even when CN aplasia was documented. For instance, a subset of children with CHARGE syndrome benefits from CI because the residual CN fibers exist, although they are very small and follow an alternative course. Additionally, children with apparent CN aplasia on MRI could benefit from electrical stimulation to develop auditory performance, which is likely due to connections or anastomoses between the CN and the adjacent vestibular nerve based on anatomical studies [18,19]. Corroborating this, our results also support that CI may be useful as an initial treatment modality before ABI in children with CN aplasia, as evidenced by the result that 57.1% (12/21) obtained a CAP score ≥3 at 3 years post-CI. 

However, ABI could be an alternative treatment modality for CI in a subset of children with CN aplasia. Our data presented herein suggest that transiting auditory rehabilitation from CI to ABI may be considered for better auditory performance, especially in cases with limited benefit at the early postoperative stage (i.e., CAP score ≤1 at 1 year postoperation). To the best of our knowledge, reproducible and reliable prognostic markers, including imaging and audiological data, that can predict CI outcomes in cases with CN aplasia are scarce. We, for the first time, suggest that changes in auditory performance at the early postoperative stage might serve as a possible prognostic marker to predict the trajectory of auditory performance postoperatively. Indeed, possible cofactors, such as demographic and surgical approaches, were well-controlled between the groups according to the CAP score at the early postoperative stage. Supporting this, Vesseur et al. [6] suggested that progress towards alternative treatment modalities such as ABI should be considered to obtain better outcomes when there is no response within several months following CI. Indeed, moderate hearing benefit after switching to ABI was observed in cases with unsuccessful CI outcomes. Furthermore, a meta-analysis of nontumor pediatric ABI [20], predominantly with CN aplasia (103/162, 64.6%), reported that 47.9% of ABI recipients achieved CAP scores >4 at 5 years postoperation [20]. Interestingly, Aslan et al. [21] showed that pediatric ABI recipients with late implantation (age at implantation ≥ 3 years) had poorer auditory performance than those with early implantation even after 5 years of ABI insertion. In other words, a delayed transition from CI to ABI, up to 3 years after CI surgery, would be inappropriate, precluding sensitive time for auditory and language development. However, further studies are required to elucidate additional evidence on the optimal time point for transiting auditory rehabilitation.

In clinical practice, intraoperative ECAP is measured to confirm electrode placement, which is correlated with the spiral ganglion neuron (SGN) population. Typically, ECAP is useful for determining the initial programming level and estimation of audiologic outcomes [22,23], and significant correlations between speech perception after CI and ECAP parameters have been documented in the literature [24,25,26]. Similarly, some authors have suggested that the absence of ECAP is associated with poor audiologic outcomes in children with CN deficiency [25,27]. However, poor responsiveness and the possibility of electrical artifacts between electrodes and CN fibers make it difficult to use ECAP in patients with CN aplasia [28]. Furthermore, previous studies have indicated that the electrically evoked auditory brainstem response elicited by CI was more sensitive than the ECAP. Yamazaki et al. showed that electrically evoked auditory brainstem response testing, coupled with CN integrity on MRI, is clinically meaningful for predicting postoperative CI outcomes. In this study, we observed that the ECAP response rate was not significantly correlated with language development after CI in children with CN aplasia, albeit with a weak relationship. Perhaps, diverse electrodes and a small cohort may have hindered the drawing of firm conclusions regarding the relationship between intraoperative ECAP and postoperative CI outcomes.

This study has some limitations that should be addressed in future studies. In particular, the retrospective nature of this study may limit the generalizability of our results. Although all cochlear implant recipients underwent comprehensive preoperative evaluation, including check-ups for significant global or neurodevelopmental delays, potential confounding factors that may affect language outcomes after CI, such as delayed developmental delay and comorbidities (e.g., medical syndromes) [5,29], were not completely evaluated in our cohort, which might have biased the results of this study. Our main findings depended on CAP score for the assessment of auditory development in subjects. Although it has relatively high interrater reliability, there is a limitation in that CAP score is a subjective test [30]. Furthermore, the lack of a control group may weaken our findings. Therefore, future case-control studies with a prospective study design, as well as matched cofactors, would have a stronger significance. Although we observed that the CAP score at 1 year postoperation may serve as a prognostic value for the longitudinal improvement of auditory performance in children with CN aplasia, further investigation is needed to support our results.

## 5. Conclusions

We elucidated the outcome of CI in the largest number of patients with bilateral CN aplasia in the literature, based on cross-sectional and longitudinal audiological analyses. Our results further refine those of previous studies on the clinical feasibility of CI as the first treatment modality to elicit favorable auditory performance in children with CN aplasia. However, this benefit may be predominant in a subset of pediatric CI recipients manifesting a CAP score >1 at the early postoperative stage. Thus, special attention should be paid to switching auditory rehabilitation from CI to ABI, especially in pediatric patients with unsuccessful cochlear implants at the early postoperative stage.

## Figures and Tables

**Figure 1 medicina-58-01474-f001:**
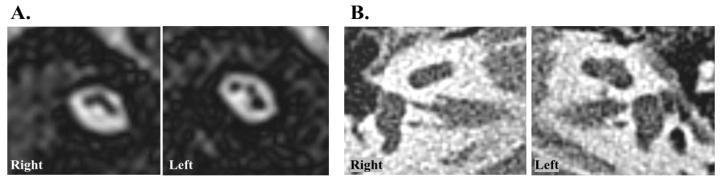
Representative imaging of cochlear nerve aplasia. (**A**) nonvisible cochlear nerve on oblique sagittal view of internal auditory canal magnetic resonance imaging. (**B**) absence of bony cochlear narrow canal on temporal bone computed tomography.

**Figure 2 medicina-58-01474-f002:**
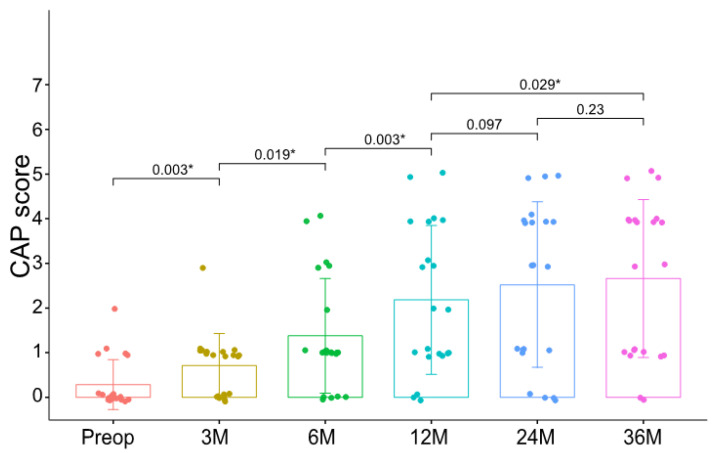
Preoperative and postoperative CAP scores in pediatric cochlear implantees with bilateral cochlear nerve deficiency. *p*-values are presented on the comparing lines. *, statistical significance (by Wilcoxon signed-rank test). CAP, Category of Auditory Performance; Preop, preoperative; M, postoperative months.

**Figure 3 medicina-58-01474-f003:**
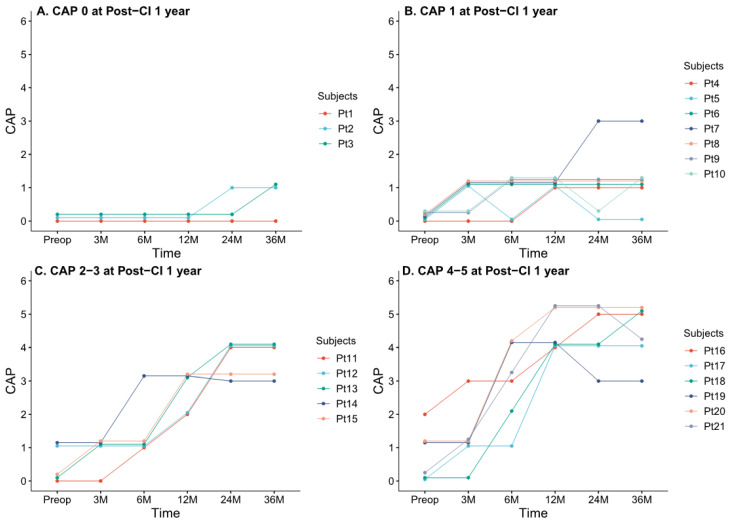
Longitudinal changes in CAP scores according to CAP score at 1 year postoperation. Patients were subclassified into four groups based on CAP scores at 1 year postoperation and depicted individual postoperative auditory performance. CAP, Category of Auditory Performance; CI, cochlear implant; Preop, preoperative; M, postoperative months.

**Figure 4 medicina-58-01474-f004:**
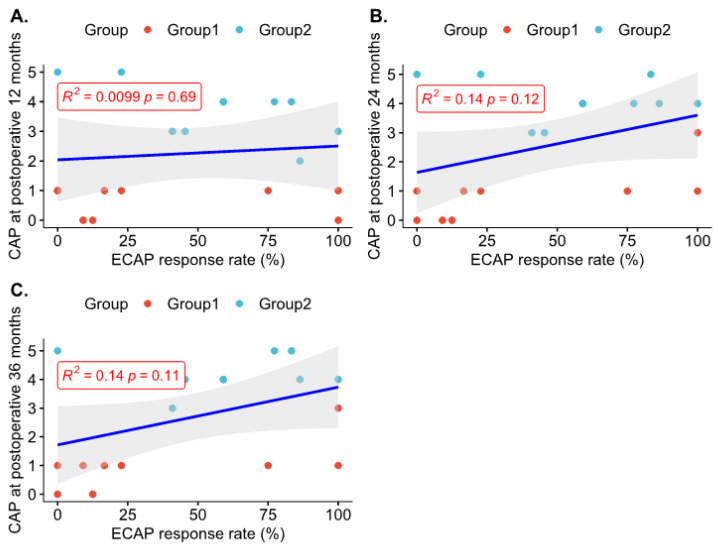
Correlation analyses between the intraoperative positive ECAP ratio and CAP scores at postoperative stages. No statistically significant correlations between intraoperative ECAP response rate and CAP scores at 1 (**A**), 2 (**B**) and 3 years (**C**) after surgery were illustrated (using Spearman’s correlation analysis). Group 1 consisted of participants with CAP scores of 0-1 at 1 year postoperation, and Group 2 consisted of participants with CAP scores ≥2 at 1 year postoperation. CAP, Category of Auditory Performance; ECAP, electrically evoked compound action potential.

**Table 1 medicina-58-01474-t001:** Demographics and clinical characteristics of pediatric cochlear implantees with cochlear nerve aplasia.

Patients (*n* = 21, 37 Ears)	
Age at CI (months)	
Mean (SD)	17.71 (6.99)
Range	10–37
Sex	
Male	10 (47.6%)
Female	11 (52.4%)
Laterality	
Unilateral, right	2 (9.5%)
Unilateral, left	3 (14.3%)
Bilateral, simultaneous	16 (76.2%)
Manufacturer	
Cochlear	15 (71.4%)
Medel	5 (23.8%)
Advanced bionics	1 (4.8%)
Approach	
Round window	16 (76.2%)
Cochleostomy	5 (23.8%)
Inner ear anomaly	
With anomaly *	8 (38.1%)
Without anomaly	13 (61.9%)

CI, cochlear implantation; SD, standard deviation. * note that various cochleovestibular anomalies combined with cochlear nerve aplasia are described in Appendix A.

**Table 2 medicina-58-01474-t002:** Comparison of clinical profiles between patients with CAP score ≤1 versus patients with CAP score >1 at 1 year postoperation.

	CAP Score ≤ 1 (*n* = 10)	CAP Score > 1 (*n* = 11)	*p*-Value
Sex (M:F)	6:4	4:7	0.519
Age (Mean ± SD, months)	17.4 ± 5.2	18 ± 8	0.94
Side (Unilateral:Bilateral)	3:7	2:9	0.903
Approach (RW:Cochleostomy)	9:1	7:4	0.366
Inner ear anomaly	4 (40%)	4 (36.4%)	1.00
CAP score at baseline	0	0.55 ± 0.66	0.02
CAP score at 3 months	0.4 ± 0.5	1 ± 0.7	0.048
CAP score at 6 months	0.5 ± 0.5	2.2 ± 1.2	0.001
CAP score at 12 months	0.7 ± 0.5	3.5 ± 1.0	<0.001
CAP score at 24 months	0.8 ± 0.9	4.1 ± 0.7	<0.001
CAP score at 36 months	1 ± 0.8	4.2 ± 0.6	<0.001
Positive ECAP ratio (%)	37.3 ± 39.6	57.4 ± 29.4	0.25

CAP, Category of Auditory Performance; ECAP, electrically evoked compound action potential; M, male; F, female; SD, standard deviation; RW, round window; ECAP, electrically evoked compound action potential.

**Table 3 medicina-58-01474-t003:** Comparison of T-level, C-level, and dynamic range at postoperative 3, 6, 12, 24, and 36 months in patients with bilateral CN aplasia.

	T-Level	C-Level	Dynamic Range
Post-CI 3 months	91.8 ± 59.8	150.4 ± 67.7	58.6 ± 17.7
Post-CI 6 months	98.5 ± 59.8	160.1 ± 59.7	61.5 ± 19
Post-CI 12 months	93.8 ± 61.9	159.6 ± 59.8	65.9 ± 24
Post-CI 24 months	88.4 ± 60.3	161 ± 57.9	72.6 ± 33.6
Post-CI 36 months	96.7 ± 63.4	173.1 ± 52.2	76.4 ± 46.7
*p*-value	0.99	0.82	0.97

ANOVA was used for the comparison of T-level, C-level, and dynamic range between evaluation times. T-level, threshold level; C-level, comfortable level; CN, cochlear nerve; CI, cochlear implant.

## Data Availability

The data presented in this study are available on request from the corresponding author.

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
