# Peer review of "Functional Outcomes of Cochlear Implantation in Children with Bilateral Cochlear Nerve Aplasia"

_medicina, 2022, doi:10.3390/medicina58101474_

Round 1

Reviewer 1 Report

Thank you for interesting paper

Anyway please correct some issuess:

1. In line 110-111 you mentioned "As outlined in the guidelines [15], test conditions for neural response telemetry were set" - I haven't found guideline in that paper

2. Line 112-113 " The maximum stimulus was set at a current level of 255 mA" ??? Too much in my oppinion - please correct it

3. Line 359: "Funding: Please add: This research received no external funding" - I guess that "please add" is not necessary

4. It would be beneficial if you compare CAP results for CN aplasia with no aplasia CI mean results.

5. 5 of 21 of your patients are unilaterally implanted. I wonder if they have worse CAP results?

Author Response

  1. In line 110-111 you mentioned "As outlined in the guidelines [15], test conditions for neural response telemetry were set" - I haven't found guideline in that paper

[Response] We appreciate you letting us know about the citation error. It has been corrected and the relevant paper has been cited.

  1. Line 112-113 " The maximum stimulus was set at a current level of 255 mA" ??? Too much in my oppinion - please correct it

[Response] Thanks for your detailed review. The unit of the current level has been changed to "CL" as in the paper which was mentioned in response 1.

“The maximum stimulus was set at a current level of 255 CL.”

  1. Line 359: "Funding: Please add: This research received no external funding" - I guess that "please add" is not necessary

[Response] Thanks for finding typo. The phrase has been deleted.

  1. It would be beneficial if you compare CAP results for CN aplasia with no aplasia CI mean results.

[Response] Thanks for your constructive comments. Of course, we share the same viewpoint as the reviewer. Previous studies have already reported significant differences between CN aplasia and others (CN hypoplasia and normal CN) in auditory performance after CI, indicating that the higher the population of spiral ganglion neurons, the better the CI outcomes. We believe the relationship between spiral ganglion neuron population and CI results is already well established. Instead, we would like to focus on functional outcomes in children with bilateral CN aplasia who underwent unilateral or bilateral CI using cross-sectional and longitudinal assessments.

  1. 5 of 21 of your patients are unilaterally implanted. I wonder if they have worse CAP results?

[Response] We appreciate your insightful feedback. According to the reviewer's judgment, we compared the mean CAP scores of 16 patients with bilateral implants and 5 patients with unilateral implants. The mean CAP scores were 3 and 2, respectively, as a result. However, the Mann-Whitney test yielded a p-value of 0.2877, indicating that no statistically significant difference existed. These findings could be attributed to the study's small sample size. I believe that comparing the results of the two groups can create new, meaningful results if the sample size is increased.

Reviewer 2 Report

The present study is clear, easy to read, well designed. Even if the study is retrospective, it aimed to find predictive factors in rehabiliation decision making in case of cochlear nerve hypoplasia or aplasia. I’m looking forward the next step with a prospective study comparing ABI / CI outcomes and evaluating ECAP and electrically evoked auditory brainstem response

Author Response

We appreciate the reviewer's thoughtful comments. The outcomes of follow-up studies on ABI, which will be the next stage in this study, are something we are also anticipating, as you noted.

Reviewer 3 Report

Dear Editor,

I reviewed the article entitled “Functional Outcomes of Cochlear Implantation in Children 2 with Bilateral Cochlear Nerve Aplasia” by Goun Choe et al.

The article is an original study discussing the outcome of cochlear implantation in patients with cochlear nerve aplasia.

The article presents a study that is well designed and extremely use for the overall management of patients with cochlear nerve aplasia.

Consequently, in my opinion, it should be published.

Just one suggestion:

Line 33: it is clinically diagnosed

Author Response

First of all, we appreciate the reviewer's precious comments and suggestion. 

Line 33: it is clinically diagnosed

[Response] Thanks for the considerate comments from the reviewer. As per the reviewer's opinion, the sentence has been modified as follows.

“Cochlear nerve (CN) deficiency refers to a small or absent CN, it is clinically diagnosed using high-resolution imaging, and was first described by Casselman in 1997.”

Reviewer 4 Report

The authors wrote a very sound paper about their retrospective investigation of the cochlear implant outcome in children with cochlear nerve aplasia.

I have only three minor suggestions for the paper, that would not mount up to minor revision:

Line 89: add the mean age of the children included in the study. Alternatively, you could refer to Table 1.

Line 115: add the unit (qu) to "charge units". Else some readers may stumble at qu/s.

Table 1: remove the line under "Age at CI (months)"

I congratulate the authors for such a fine manuscript.

Author Response

First of all, we appreciate the reviewer's precious comments and suggestions. 

Line 89: add the mean age of the children included in the study. Alternatively, you could refer to Table 1.

[Response] As per the reviewer's thoughtful comments, we add the mean age of children to Line 89.

“Ultimately, 21 patients (37 ears, mean age 17.71 months) were included in this study.”

Line 115: add the unit (qu) to "charge units". Else some readers may stumble at qu/s.

[Response] Thanks for the reviewer's thoughtful comment, we've added "qu" to the sentence.

“The maximum charge was set to 35 charge units (qu), and the charge increase rate was 8.0 qu/s.”

Table 1: remove the line under "Age at CI (months)"

[Response] Thanks for letting us know about our mistake. We deleted the line you mentioned in Table 1.
